# Tizoxanide Antiviral Activity on Dengue Virus Replication

**DOI:** 10.3390/v15030696

**Published:** 2023-03-07

**Authors:** Kristie A. Yamamoto, Kevin Blackburn, Michael B. Goshe, Dennis T. Brown, Edimilson Migoswski, Isabele B. Campanhon, Monica F. Moreira, Davis F. Ferreira, Marcia R. Soares

**Affiliations:** 1Department of Biochemistry, Institute of Chemistry, Universidade Federal do Rio de Janeiro, Rio de Janeiro 21941-909, Brazil; 2Department of Molecular and Structural Biochemistry, North Carolina State University, Raleigh, NC 27607, USA; 3Institute of Pediatrics and Puericulture Martagão Gesteira, Universidade Federal do Rio de Janeiro, Rio de Janeiro 21941-912, Brazil; 4Department of Virology, Paulo de Góes Microbiology Institute, Universidade Federal do Rio de Janeiro, Rio de Janeiro 21941-902, Brazil

**Keywords:** antiviral, tizoxanide, dengue virus, virucidal activity

## Abstract

Dengue virus is an important circulating arbovirus in Brazil responsible for high morbidity and mortality worldwide, representing a huge economic and social burden, in addition to affecting public health. In this study, the biological activity, toxicity, and antiviral activity against dengue virus type 2 (DENV-2) of tizoxanide (TIZ) was evaluated in Vero cell culture. TIZ has a broad spectrum of action in inhibiting different pathogens, including bacteria, protozoa, and viruses. Cells were infected for 1 h with DENV-2 and then treated for 24 h with different concentrations of the drug. The quantification of viral production indicated the antiviral activity of TIZ. The protein profiles in infected Vero cells treated and not treated with TIZ were analyzed using the label-free quantitative proteomic approach. TIZ was able to inhibit virus replication mainly intracellularly after DENV-2 penetration and before the complete replication of the viral genome. Additionally, the study of the protein profile of infected not-treated and infected-treated Vero cells showed that TIZ interferes with cellular processes such as intracellular trafficking and vesicle-mediated transport and post-translational modifications when added after infection. Our results also point to the activation of immune response genes that would eventually lead to a decrease of DENV-2 production. TIZ is a promising therapeutic molecule for the treatment of DENV-2 infections.

## 1. Introduction

In Brazil, the focus of attention towards COVID-19 ended up weakening surveillance actions of endemic diseases and viruses carried by arthropods, which were neglected. As a consequence, the main cause of diseases such as dengue, Zika, and chikungunya (the *Aedes aegypti* mosquito) [1] have returned with a strong impact to the country. In the begging of 2022 alone, a total of 161,605 notifications of probable infections were registered between 2 January and 12 March. According to the Ministry of Health epidemiological bulletin, the incidence is 75.8 per 100,000 inhabitants. In fact, there were 154 severe cases of dengue and 1504 cases with warning signs and symptoms, such as severe abdominal pain, persistent vomiting, and bleeding gums.

Despite 70 years of study, dengue remains a global health threat. The increase in the world population, the increase in densely populated areas, and the modernization of public transportation facilitate the spread and emergence of new infections or their maintenance in the population [2], such as the COVID-19 pandemic.

In this context, a greater knowledge on the virus nature and its replication mechanisms and also on non-toxic antivirals such as TIZ are important tools for fighting each virus threat more efficiently.

The object of study in the present work is the dengue virus type 2 (DENV-2), due to its public health impact in several countries [3]. A total of 3,126,184 cases of arboviral disease were reported in 2022. Of those, 2,812,250 (90.0%) were dengue cases (https://www.paho.org/plisa, accessed on 17 February 2023). DENV has four distinct serotypes (DENV-1, DENV-2, DENV-3, and DENV-4) that are transmitted to humans by *Aedes aegypti* or *Aedes albopictus* mosquitoes [4]. Most infections by any of the serotypes can produce a spectrum of symptoms that range from mild cases and classic and self-limited dengue fever (DF) to more severe cases such as hemorrhagic fever (DHF) and dengue shock syndrome (DSS), although most are asymptomatic [5]. Although all four dengue virus serotypes are endemic, DENV1 and DENV2 were the most prevalent in the Americas Region in 2021 and 2022.

The development of dengue vaccines began in the 1920s, but it is challenging to build immunity against all four dengue serotypes. Currently, there is a commercial version known as CYD-TDV, sold under the brand name Dengvaxia [6]. However, the vaccine is recommended only for those who have already had dengue or populations in which most people have been infected previously.

The search for antiviral compounds began in the 1950s when sulfonamide antibiotics were tested in the first in vitro antiviral trials using embryonated eggs [7]. As of 2016, about 90 antivirals are available for clinical use [8]. Most of these drugs were introduced in the last 20 years, being used to treat patients infected with HIV, hepatitis B and C, VZV, cytomegalovirus (CMV), respiratory syncytial virus (RSV), and influenza [9,10]. It should be noted that, to date, no antiviral drug can cure a dengue virus patient; treatment addresses symptoms only. Antivirals used to control several virus infections have also been linked to the selection of resistant mutants, making it even more difficult to fight and control viral infections [11]. In this way, the emergence and re-emergence of new viruses and the development of viral resistance have encouraged the search for new broad spectrum antiviral agents. The development of compounds against DENV has proven promising. Recently, some studies aimed to discover new antivirals, either from the synthesis of new compounds or extracts from plants and bacteria [12,13,14,15,16]. 

This work investigated the activity of tizoxanide (TIZ) against DENV-2. TIZ is the active ingredient in nitazoxanide (NTZ), clinically approved antiparasitic agent widely marketed in the United States, India, and Latin America [17]. In addition, NTZ and its active metabolite, TIZ, have been shown to inhibit a range of gram-positive and gram-negative bacteria and the replication of several DNA and RNA viruses [18,19]. In a recent study [20], it was shown that the treatment of Vero cells with TIZ causes a small change in the protein profile of cells, decreasing protein expression, mainly related to protein translation and linked processes to RNA, consistent with a decrease in cellular metabolism. In another study, Rocco et al. [21] reported that early nitazoxanide therapy was safe and decreased viral load significantly in SARS-CoV-2 patients. 

In this study, different assays were carried out to investigate the antiviral action of TIZ, including dose–response, virucidal, and antiviral activities. The proteome of Vero cells infected and treated with TIZ for 72 h was also analyzed. The results showed significant differences in protein expression levels in response to TIZ treatment that may be related to the viral inhibition process.

## 2. Materials and Methods

### 2.1. Reagents

Tizoxanide (C10H7N3O4S; MW 265.25; catalog #SC-208441; Santa Cruz Biotechnology Inc., Rio de Janeiro, Brazil) was provided by Farmoquímica S/A. The substance was solubilized in dimethyl sulfoxide (DMSO), stored at −20 °C, and the final dilutions for the experiments were carried out in DMEM culture medium (Dulbecco’s Modified Eagle Maintenance Medium). Ribavirin (1-β-D-Ribofuranosyl-1,2,4-triazole-3-carboxamide; C8H12N4O5; MW 244.20) is a guanosine analogue with broad antiviral activity. Ribavirin (Sigma-Aldrich, St. Louis, MO, USA) was used as a positive control for DENV-2 inhibition [22].

### 2.2. Cell Culture

Vero cells (African green monkey kidney cells, ATCC CCL-81) were maintained at 37 °C with 5% CO_2_ in Dulbecco’s modified Eagle maintenance medium (DMEM) supplemented with 10% fetal bovine serum (Gibco-Life Technologies, Carlsbad, CA, USA), 2 mM L-glutamine, 10 mM HEPES (pH 7.4) and non-essential amino acids (1:100 dilution of non-essential amino acids, cat. No. 11140, Gibco, Carlsbad, CA, USA). For the assays, 3 × 10^5^ cells/mL were cultured in cell culture dishes or bottles for 24 h, to reach confluence, by trypsinization of the confluent culture.

### 2.3. Determination of Viremia and Protein Concentration in Dengue-Virus-Infected Cells

DENV serotype 2 (New Guinea C strain) was adsorbed to cells with a multiplicity of infection equal to 0.02 for 1 h to prepare the viral stock. After this time, the medium containing the non-adsorbed virus was replaced by cell culture medium and incubated for up to 168 h at 37 °C. At this time, the cells were with about a 70% evident cytopathic effect. The supernatant was collected and clarified by centrifugation at 3000 rpm to remove cellular debris. The supernatant was then aliquoted and placed at −70 °C with 10% glycerol until use. The degree of purity of the viral preparation was evaluated by electrophoresis in 15% polyacrylamide gel containing sodium dodecyl sulfate (SDS-PAGE) [23], where only the structural viral proteins should be observed. The concentration of viral proteins was determined using a Lowry-based colorimetric assay of the DC Protein Assay kit (Bio-Rad Laboratories Inc., Hercules, CA, USA), according to the manufacturer’s recommendations, and the calculation of total virus particles (infectious and non-infectious) was performed considering a protein mass of 5.5 × 10^–11^ μg per virion. The physical integrity and structural homogeneity of the purified viral particles were evaluated by negative staining using transmission electron microscopy (described below). The titer of the virus preparation was determined by plaque assay.

### 2.4. Viral Titration by Plaque Assay

The titer of viral samples was determined by plaque assays [24]. Briefly, monolayers of Vero cells, grown in 6-well plates (TPP), were inoculated with 200 μL of serial dilutions of the viral sample for 1 h. After viral adsorption, the inoculum containing the non-adsorbed viruses was removed and replaced with semi-solid culture medium (2% carboxymethyl cellulose in maintenance medium). After 168 h (DENV) incubation at 37 °C, cells were fixed with 20% formaldehyde in PBS and stained with 1% crystal violet to determine the number of plaques. The viral titer was calculated from the number of PFU obtained in a given well, multiplied by the dilution factor, expressed in PFU/mL.

### 2.5. Pre-Treatment Test

The assay of the influence of pretreatment of cells with TIZ was determined according to Yang et al. [25], with modifications. The monolayers of Vero cells grown in 6-well plates were treated with varying concentrations of the substance for 24, 12, or 1 h at 37 °C. Afterwards, the cells were washed with PBS and infected with about 300 PFUs per well for 1 h for viral adsorption. The culture medium containing the non-adsorbed viruses was then replaced with semi-solid medium for plaque assay. DENV was incubated for 168 h and fixation and staining was performed as previously described for titration. The %VI was calculated similarly to the dose–response curve. This assay and the other assays of the antiviral effect were carried out in duplicate, in three independent experiments.

### 2.6. Assay of Inhibition of Virus Adsorption

To evaluate the effect of TIZ on virus adsorption, cultured Vero cells grown in 6-well plates were previously cooled for 10 min at 4 °C according to modifications in Zhu et al. [26]. Then, the monolayer was treated with varying concentrations of TIZ for 30 min, also at 4 °C. The cells were then rinsed with PBS at 4 °C and infected with 300 PFUs at 4 °C for virus adsorption. After 1 h, the monolayer was rinsed again with PBS at 4 °C for the removal of the non-adsorbed virus and a subsequent plaque assay was performed. The cells were incubated for 168 h at 37 °C.

### 2.7. Virus Penetration Inhibition Assay

The assay was performed according to Su et al. [27], for evaluating the effect of TIZ on the penetration of the virus. The cells grown in 6-well plates were previously cooled at 4 °C for 10 min, followed by infection with 600 PFUs for 1 h at 4 °C. The cells were then rinsed with ice-cold PBS and different concentrations of cooled TIZ were added to the monolayer and incubated at 37 °C. After 10 min of treatment, the cells were rinsed with acidic PBS (pH 3) for 1 min to inactivate the non-penetrated viruses, and then, the semi-solid medium was added to perform the plaque assay for 168 h.

### 2.8. Virucidal Activity

The direct effect of the substance on the virus particle was evaluated according to Cheng et al. [28], with modifications. The stock virus suspension was pre-incubated with TIZ (*v*/*v*) for 1 h at 37 °C. The treated virus sample was then diluted and inoculated into cells for 1 h for viral adsorption, and residual infectivity was determined by plaque assay. The dilution of the sample significantly reduces the concentration of the drug that is incubated with the cells (dilution of at least 100×). Thus, it is estimated that the reduction in the titer is due only to the inactivation of the virus, without the presence of the drug in an effective concentration.

### 2.9. Post-Infection Assay

Confluent cell cultures grown in 6-well plates were infected with 300 PFUs of DENV-2 for 1 h at 37 °C. After removal of non-adsorbed virus and rinsing of the cell monolayer, the cells were treated with the substances from 0 to 3 h or from 3 to 6 h after virus infection (hpi). The cells were then rinsed with PBS to remove excess substance, and semi-solid medium was added to the culture.

### 2.10. Dose–Response Inhibition Curve

The dose–response inhibition curve was made according to Santos et al. [29], using the virus production inhibition assay. Confluent cells grown in 24-well plates were infected with MOI = 0.02 (DENV) for 1 h at 37 °C. The inoculum was then removed, the cells were rinsed with PBS, and different concentrations of TIZ diluted in DMEM were added. After 72 h (DENV) incubation at 37 °C, the supernatant containing the viruses produced and released by the cells during this time was collected and titrated. Infected and untreated cells were considered as virus control, as 100% viral replication; uninfected and untreated cells were considered as cellular control; and cells infected and treated with ribavirin were used as the positive control of viral replication inhibition. 

The percentage of viral inhibition (%VI) was determined by the formula: %VI = (1 − (mean of the number of PFU in the treatment/average of the number of PFU in the viral control)) × 100. The concentration of the substance capable of inhibiting 50% and 90% of viral replication (IC50% and IC90%) was determined by linear regression analysis of the dose–response inhibition curve for each substance. The selectivity index (IS) was calculated by the ratio between CC50% and IC50%. The relative potency (PR) was calculated by the theoretical potential formula, expressed by the ratio between the IC90% of the reference substance (ribavirin) and the IC90% of the substance tested. Assays were performed in quadruplicate, and three independent experiments were performed.

### 2.11. Virus Release Assay

Virus release assay was performed according to Su et al. [27] and similarly to the dose–response curve. Cells grown in 24-well plates were infected for 1 h for viral adsorption. Then, the inoculum was removed, the cells were rinsed, and the TIZ was added. After 72 h of incubation, the supernatant was collected, the cells were rinsed, and the same volume of maintenance medium as the supernatant was added to the monolayer. The cells were then disrupted by freezing and thawing (3×), and viral production of the supernatant and monolayer was determined by plaque assay.

### 2.12. Real-Time Polymerase Chain Reaction via One-Step Reverse Transcription

Real-time RT-PCR was performed with the virus release assay sample. After the incubation of infected and/or treated cells for 72 h, viral RNA was extracted from the supernatant or cell monolayer with the QIAamp^®^ Viral RNA mini kit (QIAGEN, Hilden, Germany), according to the manufacturer’s recommendations. Each RNA sample was diluted 10, 100, and 1000 times, and only one assay was performed. Real-time RT-PCR for DENV was conducted in a 96-well plate using the FTD Dengue Differentiation Kit (Fast-track Diagnostics, Siemens Healthcare, Luxembourg, BE) and performed according to the manufacturer’s recommendations. Briefly, viral RNA was transcribed into cDNA via reverse transcription using a specific primer, followed by the polymerase chain reaction in the same well. The specific presence of DENV sequences in the reaction was detected by the increase in fluorescence observed from the doubly-labeled probe, and was reported as Ct values by the PikoReal™ Real-Time PCR System thermocycler and analyzed by PikoReal Software 2.1 (Thermo Scientific, Waltham, MA, USA). The total reaction volume was 12.5 μL, including 5 μL of sample, 0.75 μL of probe mix and primers from the FTD Dengue differentiation kit, and 0.5 μL of enzyme and 6.25 μL of AgPath buffer-ID™ One-Step RT-PCR Kit (Life Technologies, Carlsbad, CA, USA). The thermal cycler was programmed for an initial incubation of 50 °C for 15 min, followed by a second incubation of 95 °C for 10 min and 45 cycles of 10 s at 95 °C and 40 s at 60 °C. The Ct values obtained were compared with the viral titer of the samples. Thus, the lower the viral load, the lower the fluorescence emitted and, consequently, the higher the Ct of the sample. The experiments were performed in triplicate.

### 2.13. Statistical Analysis

All assays were performed in duplicate, triplicate, or quadruplicate, and three independent experiments were conducted. The substance-treated and virus-infected samples from each assay were compared to an untreated infected control. All variables were submitted to analysis of variance (ANOVA), followed by a Tukey’s test or *t*-test through the JMP^®^ program. Data were represented as mean ± standard deviation and *p*-values < 0.05 were considered statistically significant. 

### 2.14. Protein Digestion to Proteomics Analysis

The filter-aided sample preparation (FASP) method was used for the purification and on-filter digestion of proteins, based on Wiśniewski et al. [30]. All buffer exchanges were carried out by centrifugation at 10,000× *g* for 15 min. An amount of 200 μg of protein extract was reduced with a final concentration of 5 mM DTT at 56 °C for 30 min and transferred into a 500 μL Sartorius Vivacon 30,000 MWCO centrifugal unit (Fisher Scientific, Rockford, IL, USA). Then, 200 μL of UA buffer (8 M urea, 0.1 M Tris, pH 8.5) was added and the sample was centrifuged (step repeated once). For the alkylation of reduced cysteine bonds, 100 μL of UA buffer containing 50 mM iodoacetamide was added and incubated in the dark for 20 min at room temperature following centrifugation. Three 100 μL UA buffer exchanges were used to remove residual alkylating agent, followed by three buffer exchanges with 100 μL of ABC buffer (50 mM ammonium bicarbonate buffer, pH 8). A volume of 50 μL of ABC buffer containing a 1:100 ratio of sequencing-grade modified trypsin (Promega, Madison, WI, USA) to protein was added, and the tubes were incubated at 37 °C in a water bath for 18 h. Two rounds of 40 μL of ABC buffer were used to elute the peptide-rich solution, and then, the samples were dried down using vacuum centrifugation. Three biological samples for each condition were prepared in this manner and used for sequence analysis by LC-MS/MS. 

### 2.15. Liquid Chromatography-Tandem Mass Spectrometry (LC/MS/MS) Analysis

Tryptic digests of each sample obtained by the FASP method were analyzed by LC/MS/MS using an Easy NanoLC 1000 (Thermo Scientific, Waltham, MA, USA) coupled to an Orbitrap Elite mass spectrometer (Thermo Scientific). Digests were desalted and preconcentrated onto a 2 cm × 100 µm i.d. Pepmap C18 (5 µm particle size) (Thermo Scientific), and then eluted onto and separated using a self-packed PicoFrit (New Objective, Woburn, MA, USA) 75 µm id × 25 cm Magic C18 column (3 µm particle size) with a 60 min linear gradient from 2% mobile phase B to 40% mobile phase B (A = 2% acetonitrile in water, 0.1% formic acid; B = acetonitrile, 0.1% formic acid). An electrospray voltage of 2.8 kV was applied to the PicoFrit column to ionize peptides in the nanoelectrospray ion source of the Obitrap Elite with a heated capillary temperature of 275 °C. The data acquisitions of MS (scan range of *m*/*z* 400–2000) and MS/MS (scan range of *m*/*z* 140–2000) were collected utilizing the Orbitrap analyzer. A top-five method with higher-energy collisional activation (HCD) for product ion generation in the HCD cell was used with a normalized collision energy setting of 27 V to induce precursor ion fragmentation (+1 charge states were excluded). 

### 2.16. Protein Identification and Analysis 

Raw LC/MS/MS datafiles were processed using MaxQuant software [31] and database-searched using the integrated Andromeda [32] search engine against the non-redundant database of the National Center of Biotechnology Information (NCBI) containing entries from Chlorocebus sabaeus, as well as the two protein standards bovine serum albumin and rabbit glycogen phosphorylase (62,148 entries). Trypsin was defined as the digesting enzyme, along with a maximum of two missed tryptic sites, whereas fixed carbamidomethyl Cys modification and variable oxidized Met modification were permitted. A false discovery rate (FDR) of 0.05 was utilized. Otherwise, default MaxQuant and Andromeda parameters were used for processing and searching. For statistical analysis, the MaxQuant ProteinGroups report was imported into Perseus, and protein label free quantification (LFQ) intensities based on extracted ion chromatograms were used to compare expression differences between sample groups. Data preprocessing prior to analysis of variance (ANOVA) included the following steps: log (2) transformation of protein intensities and replacing missing data with a value approximating the lower limit of detection. Only proteins that were observed in three out of three replicates for at least one treatment group were retained and subjected to ANOVA. Statistically significant proteins were retained at the *p* < 0.05 significance level.

## 3. Results

### 3.1. Antiviral Activity

The first assay was performed to determine the protective effect of TIZ, to examine whether TIZ could protect cells from viral infection. For this, Vero cells were treated with increasing concentrations of TIZ for 1, 12, or 24 h before infection, as described in Methods. The results show that TIZ was not able to protect Vero Cells for DENV-2 infection (Appendix A).

### 3.2. Antiviral Effects against DENV-2

Viral adsorption inhibition by TIZ was evaluated. The following assay was performed, TIZ was added to the cell monolayer at 4 °C, and after 30 min, the cells were infected with DENV-2. The results show that TIZ treatment did not decrease the number of PFU compared to the control assays, not interfering with the DENV-2 adsorption to Vero cells (Appendix A). 

To observe the effect of TIZ on DENV-2 penetration, Vero cells were infected for 1 h at 4 °C to allow viral adsorption. The cells were then treated with varying concentrations of TIZ and incubated for 10 min at 37 °C to allow the virus to penetrate the cell membranes. To inactivate the viruses that did not penetrate, the culture was rinsed with acidic PBS for subsequent performance of the plaque assay. The results show that the number of PFU in the treatment and in the control was not significantly altered, proving the inefficiency of TIZ in preventing the penetration of DENV-2 into Vero cells (Appendix A).

### 3.3. TIZ Inhibits Viral Infection

Infected cells were treated with non-toxic concentrations of TIZ (0.5–2.2 µM) (Appendix A) and maintained throughout the incubation period. The viruses released by the cells were collected and titrated and show that the inhibition of DENV-2 is directly proportional to the concentration of TIZ used in the treatment (Figure 1). Through linear regression, the IC50, IC90, and the selectivity index (SI) of the TIZ were determined, with values of 1.38 μM and 4.8 μM, respectively, as can be seen in Table 1. In addition, it is also possible to observe in the table below the values for ribavirin, which was used as a reference of an antiviral drug with a broad spectrum of action. Table 1 also shows the TIZ concentrations for 50% and 90% cell viability test (CC50 and CC90, respectively).

### 3.4. Loss of Infectivity of the Viral Particles by TIZ

To assay the virucidal potential, the DENV-2 stock was treated with different TIZ concentrations for 1 h at 37 °C for subsequent titration by plaque assay. This assay showed that treatment with TIZ decreases by 20% virus titer, compared to the untreated control virus titer, indicating that TIZ has a discreet virucidal action against DENV-2 (Figure 2).

### 3.5. Infectivity of Viral Particles and RT-PCR

Virus release assay was carried out to verify whether TIZ is able to influence the release of virus particles by Vero cells and whether TIZ could be influencing the infectivity of the released viral particles. For this, 72 h of TIZ treatment of cells after viral infection was followed by titration by plaque assay, and genetic viral material was evaluated by a real-time RT-PCR technique. Vero cells were infected with DENV-2 and then incubated with varying TIZ concentrations for 72 h. The viruses produced by the cells and released into the supernatant were collected and titrated, as well as the viruses that were retained within the Vero cells. From these same samples, viral RNA was also extracted to perform real-time RT-PCR for comparison with the control virus titers. The Ct values of each sample were consistent with their dilutions (neat, 1:10, 1:100, 1:1000).

The results presented in Table 2 confirm that TIZ inhibited DENV-2 production in a dose-dependent manner in both the virus that was released and the virus within cells. The percentage of virus titer inhibition from supernatant followed the same pattern of virus titer inhibition from viruses that had not been released. The inhibition for both virus conditions was TIZ-dose-dependent. The viral reduction in supernatant and cell monolayer samples were also confirmed by increasing number of Ct in RT-PCR assay, indicating that TIZ was effective in inhibiting DENV-2 viral activity in the supernatant and cell monolayer (Figure 2). The data, taken together, suggest an inhibition at the beginning of the DENV-2 infection, since that TIZ treatment does not appear to cause DEN-2 RNA to accumulate within cells, retain infectious viral particles, or prevent their release from the cell.

### 3.6. Proteomics Analysis

For quantitative analysis of the expression level of the proteins, digestion in solution was performed using the FASP protocol described in the Methods section. In one set of the 850 proteins validated by MaxQuant/Andromeda-Perseus, 22 were differentially expressed, according to statistical analysis by Perseus at the *p* < 0.05 level. Twelve of them had higher abundance in DENV-2-infected Vero cells, and ten proteins increased expression with TIZ treatment after viral infection. These proteins are involved in different biological processes such as translation, ribosomal structure and biogenesis, intracellular trafficking, secretion and vesicular transport, RNA processing and modification, and cytoskeleton (Table 3). 

## 4. Discussion

To date, there is only one licensed vaccine, which is considered unsafe, and no antiviral drug available for clinical use for dengue disease [33]. Another important highlight is that DENV-2 is the most common among the serotypes and has been a determinant factor for the emergence of dengue in different global regions [3] and epidemics with a high number of cases with warning signs [34]. Thus, the search for new chemotherapeutic agents has great importance in the context of treating DENV disease. This paper shows that TIZ has DENV-2 antiviral activity in concentrations that are non-toxic for Vero cells.

TIZ antiviral activity was confirmed by the dose–response curve, and also, antiviral assays were carried out in the different stages of viral replication to identify the most likely stage of TIZ action. The adsorption of a virus to the surface of a susceptible cell is the first step for the initiation of an infection. Then, the virus uptake into a cell involves the adsorption of viral particles and, as a second step, the penetration in which virus nucleic acid or nucleoprotein is transferred to the cytoplasmic side [35]. In this paper, Vero cells pre-treated with TIZ showed no changes in the adsorption and penetration rates of the virus into the cell when compared to non-treated cells, suggesting that the substance is not able to protect the cell from DENV-2 infection. 

In addition, low virucidal activity (Figure 2) was demonstrated for TIZ (up to 20% at the highest concentration tested), not enough to explain the high inhibition of DENV-2 replication observed. Therefore, TIZ appears to act after DENV-2 penetration into Vero cells. The inhibition of viral activity by TIZ was observed in the reduction of PFU/mL values in accordance with the increase of CT value in RT-PCR (Table 2). These data are corroborated by the subsequent studies. Virus release assay was performed together with the RT-PCR in real time, confirming that the anti-DENV effect caused by TIZ is mainly during the intracellular replication steps, after virus penetration, and minimizing the possibility of TIZ being (1) capable of accumulating viral RNA within the cell; (2) withholding the release of mature viruses; or (3) producing and releasing defective, non-infectious virus particles. 

Thus, the results suggest that TIZ acts inside the cell, before the complete replication of the viral genome, consistent with studies with other flaviviruses. Previous studies showed that NTZ, the precursor of TIZ, was not able to affect viral infectivity, adsorption, or entry of influenza and rotavirus viruses into target cells, being more effective when cells were treated after viral infection [36,37,38]. Shi et al. [39] suggested that NTZ exerts its antiviral activity against Japanese encephalitis virus (JEV) in the early stages of viral replication, after the infection of BHK-21 cells, in addition to demonstrating this inhibition in vivo, proposing a potential application of NTZ in the treatment of this virus.

Elazar et al. [40] showed that TIZ activates protein kinase R (PKR), which, in turn, results in the phosphorylation of the eukaryotic translation initiation factor 2 alpha (eIF2-α), responsible for blocking HCV replication. PKR is an important cellular enzyme in the defense against viral infections and is generally activated by double-stranded RNA, formed during the replication of several RNA viruses, by a mechanism involving autophosphorylation. Once activated, PKR phosphorylates the α subunit of eIF2, thus inhibiting protein translation and, consequently, viral replication [41]. A more recent study with another flavivirus, bovine viral diarrhea virus (BVDV), showed that NTZ treatment inhibits BVDV replication, and appears to also involve the mechanism of PKR and eIF2-α phosphorylation [42].

Recent work from our group showed that TIZ acts to reduce the overall Vero cell metabolism, which would also explain a reduction in virus production in treated cells [20].

As antiviral assays suggest that the antiviral effect exerted by TIZ occurs after the virus enters the cell, proteomic analysis was performed with the objective of comparing the proteome of the infected cell and the proteome of the infected and treated cell, to observe the differential expression of proteins that may be involved in the TIZ anti-DENV-2 response. In this study, an unlabeled mass spectrometry-based proteomics approach was used for comparative analysis between TIZ-treated and untreated Vero cells, where the protein extract was hydrolyzed in solution using filter-assisted sample preparation (FASP).

Quantitative analysis by the FASP protocol highlighted 22 differentially expressed proteins, 10 of which had their abundance decreased with treatment, and 12 of which had their abundance increased. They are involved in different biological processes, with emphasis on intracellular trafficking, secretion, and vesicular transport; and post-translational modification, protein turnover, and chaperones, consistent with the intense viral replication occurring within the cell, since many positive-polarity RNA viruses, including flaviviruses, cause the rearrangement of the host’s intracellular membrane compartments, such as ER, trans-Golgi or lysosomes, sites of post-translational modifications, and replication complexes [43,44,45]. Thus, alterations in these pathways could directly or indirectly influence DENV-2 replication.

Among the proteins whose expression was increased with treatment, sarcoplasmic/endoplasmic reticulum calcium ATPase (SERCA2) stands out. SERCA2 is an enzyme that transfers Ca^2+^ from the cytosol to the lumen of the endo/sarcoplasmic reticulum at the expense of ATP hydrolysis [46], and is essential for the dynamic balance of calcium homeostasis [47]. Ashiru et al. [42] showed that NTZ depletes the intracellular Ca2+ stores of the ER of MDBK cells, resulting in a mild ER stress (ER), which would lead to disturbances in trafficking and N-glycosylation of BVDV proteins, aiding in the antiviral effect. Calcium depletion in the ER, as well as the accumulation of misfolded proteins or protein aggregates, can activate ERS [48]. An ERS response pathway, called the UPR, is then activated [49], leading to an ER-to-nuclear signaling cascade, which is characterized by an increased expression of chaperones and other ER proteins, including SERCA2 [50].

Another protein positively expressed with the treatment was ERGIC-53, a type I transmembrane protein with a calcium-dependent and pH-sensitive carbohydrate recognition domain. ERGIC-53 acts primarily as a receptor for glycoproteins and transports them from the ER, via ERGIC, to the Golgi, reviewed by Zhang et al. [51]. In mammalian cells, ERGIC is a tube-vesicular membrane cluster system located between the RER and Golgi [52]. Qin et al. [53] showed that ERGIC-53 mRNA levels are increased in HeLa cell culture under ERS. They also showed that these proteins, which are distributed between the ER and Golgi under normal conditions, were located mainly in the Golgi complex, suggesting that under ERS, this change in location could help to prevent misfolded or misfolded proteins to be transported from the ER to the Golgi. Thus, the overexpression of ERGIC-53 would be consistent with the activation of ERS and could help to interrupt the transport of proteins, delaying the viral replication process.

Some enveloped viruses fuse their membrane with the host cell membrane, releasing their genome into the cytoplasm to initiate the viral replication cycle. In each case, one or a small group of transmembrane glycoproteins on the viral surface mediates fusion [54]. Klaus et al. [55] showed that ERGIC-53 has a conserved interaction with glycoproteins encoded by RNA viruses of several families and is essential for the formation of infectious arenaviruses and coronaviruses and filovirus particles (class I viral fusion proteins). The flavivirus E envelope glycoprotein is classified as a class II viral fusion protein [56] and there are no studies that show the positive or negative influence of ERGIC-53 on DENV fusion; as such, in this case, this protein could be acting only in the transport of these glycoproteins. It is worth mentioning that dengue viruses do not fuse their membrane with the plasma membrane during entry.

The ubiquitin-conjugating enzyme E2 N (UBE2N or Ubc13) is a member of the family of ubiquitin-conjugating enzymes E2, which mediates the synthesis of polyubiquitin chains linked to lysine 63 [57] and also had its expression increased with infection and treatment in this study. The lysine 63-linked polyubiquitin chain does not mediate protein degradation by the 26S proteasome [58]. Among other functions, together with its cofactors Uev1A or Mms2, UBE2N upregulates NF-κB activation or DNA repair, respectively [59]. When activated, the NF-κB transcription factor is released and directed to the nucleus, where it activates a range of genes essential to the immune and inflammatory response, reviewed by TERZIC et al., 2007 [60], which may help fight the infection caused by DENV-2.

Among the proteins with reduced expression with treatment, the theta subunit of the AP-2 complex and the phosphatidylinositol-linked clathrin assembly protein (PICALM/CALM) stand out. The AP-2 complex is a member of the adapter protein complex (AP complex) family, involved in intracellular transport processes, and connects transmembrane proteins to clathrin, forming the clathrin-coated vesicles responsible for the clathrin-dependent endocytosis pathway. The entry of dengue virus and other flaviviruses into the cell of mammals and mosquitoes has been described and occurs through receptor-mediated endocytosis via clathrin-coated vesicles [61,62,63,64,65], and the decrease in the expression of clathrin adapter proteins by TIZ treatment suggests an inhibition of viral replication through the inhibition of events related to virus entry and nucleocapsid release virus in the cell, despite the plaque assay experiments not demonstrating a significant effect of TIZ on the viral penetration step. It is worth remembering that in Vero cells, the internalization of DENV-2 is independent of clathrin, caveola, or lipid rafts, and dependent on dynamin [66].

In general, the data suggest that TIZ could be acting in multiple steps of the initial replication events of DENV-2, after the penetration of the virus in the cell and in later stages during protein trafficking and, consequently, virus morphogenesis. TIZ influences several cellular processes related to intracellular trafficking, post-translational modifications, and activation of the UPR response pathway to ERS and immune response genes, among others, which end up reducing the production of DENV-2 by Vero cells. As viruses are totally dependent on the cellular biochemical apparatus to replicate, any change in host cell metabolism can affect virus production. More experiments are still needed to point out the mechanisms of action of TIZ on DENV, but the results showed that the proteomics methodology seems to be a useful technology to have an overview of the cellular proteome and better understand the viral infection, in addition to helping to elucidate the cellular modifications that would be influencing the antiviral effect of substances. 

## 5. Conclusions

TIZ shows antiviral activity against DENV-2 in Vero cell culture, acting mainly in the intracellular steps of viral replication, after DENV-2 penetration in the cell and before the complete replication of the viral genome. The decrease in FASN expression by the treatment of Vero cells with TIZ suggests its involvement in the antiviral activity of this substance against enveloped viruses. The treatment of Vero cells with TIZ after infection with DENV-2 influences different cellular processes such as intracellular trafficking and post-translational modifications, in addition to seeming to activate the UPR pathway and immune response genes, which ends up culminating in a decrease in the production of these viruses by the cell. TIZ possessing antiviral activity against DENV-2 could possibly also affect other viruses. Since TIZ is able to inhibit DENV-2 and several other viruses, this molecule appears to be a promising antiviral substance with a broad spectrum of action.

## Figures and Tables

**Figure 1 viruses-15-00696-f001:**
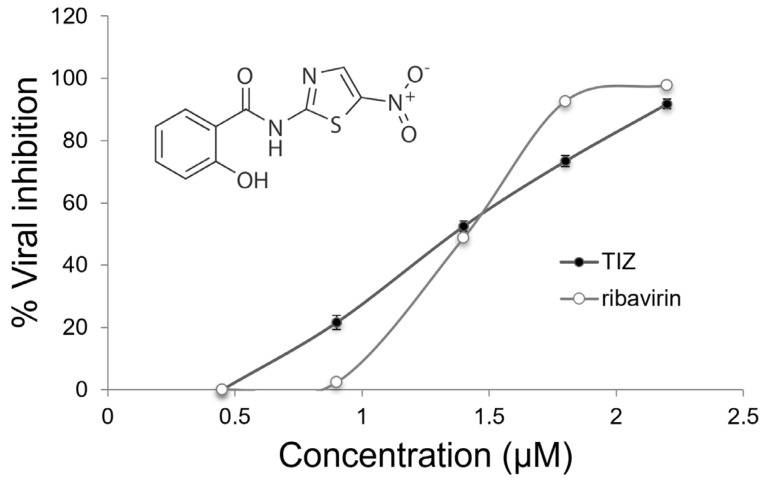
TIZ dose-response inhibition curve on DENV-2 replication in Vero cells. Different concentrations of TIZ were added to Vero cells for 72 h, right after viral infection. Then, the viral yield was titrated by plaque assay and the values were converted to %VI. The experiment shows the mean of three determinations of viral inhibition in plaque assay, compared to non-treated cell controls ± SD. The SD bars are obscured by the dot. Chemical structure of TIZ in the insert.

**Figure 2 viruses-15-00696-f002:**
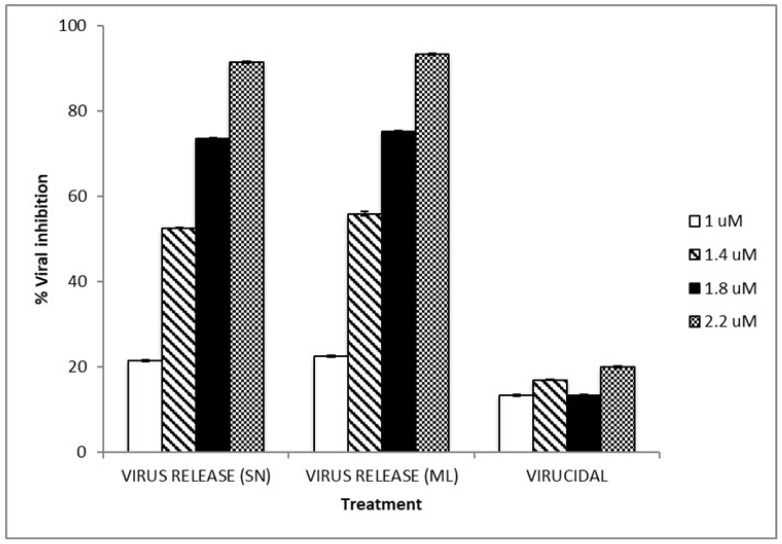
DENV-2 stock was treated with different concentrations of TIZ for 1 h at 37 °C for subsequent titration by plaque assay for virucidal test and viral release assays. The experiment shows the mean of three determinations of viral inhibition + SD in plaque assay: virus released in supernatant (SN), virus released from cell monolayer (ML), and virucidal activity.

**Table 1 viruses-15-00696-t001:** Cytotoxicity and anti-DENV-2 activity of TIZ and ribavirin.

	CC_50_ ^a^	CC_90_ ^a^	IC_50_ ^b^	IC_90_ ^b^	SI_50_ ^c^
TIZ	6.67 ± 0.30	2.63 ± 0.11	1.38 ± 0.04	2.11 ± 0.07	4.8
ribavirin	79692 ± 2390	1601 ± 40	409 ± 14	819 ± 31	194.6

^a^ Concentration for 50% and 90% cell viability in μM. ^b^ Concentration for 50% and 90% inhibition of virus production in μM. ^c^ Selectivity index = CC_50_/IC_50._

**Table 2 viruses-15-00696-t002:** TIZ antiviral activity on DENV-2 replication in Vero cell supernatant and into cells after viral infection by real-time plate and RT-PCR assay.

(uM)	PFU/mL ^a^	SD ^b^	%VI	Ct	Ct 1:10	Ct 1:100	Ct 1:1000
Supernatant							
0	1.02 × 10^6^	2.40 × 10^4^	-	16.2	18.5	22.4	25.6
1	7.97 × 10^5^	1.30 × 10^4^	21.6	16.5	18.6	22	25.6
1.4	4.83 × 10^5^	8.50 × 10^3^	52.5	16.4	18.9	22.2	25.4
1.8	2.70 × 10^5^	4.10 × 10^3^	73.4	17.1	19.8	23.2	26.3
2.2	8.33 × 10^4^	6.20 × 10^3^	91.8	18.1	21.3	24.2	27.5
Monolayers							
0	8.68 × 10^5^	1.40 × 10^4^	-	16.9	19.5	22.3	25.4
1	6.70 × 10^5^	1.50 × 10^4^	22.8	17.3	20.1	22.5	25.3
1.4	3.78 × 10^5^	2.00 × 10^4^	56.4	18.0	20.6	23.7	26.3
1.8	2.10 × 10^5^	8.20 × 10^3^	75.8	18.9	21.0	24.1	27.2
2.2	6.00 × 104	8.20 × 10^3^	93.1	20.5	22.1	25.1	28.6

^a^ Average. ^b^ Mean standard deviation.

**Table 3 viruses-15-00696-t003:** Differential protein abundance levels in DENV-2-infected Vero cells and TIZ-treated. Samples in which the proteins presented statistically significant higher expression are marked (x).

Fasta Headers	Protein Description	# Peptides	*t*-Test *p*-Value	Fold Change	Biological Process	Higher Expression	
						Control	TIZ
gi|635066390	dynactin subunit 2	3	0.012	0.2	Cytoskeleton, intracellular trafficking, secretion, and vesicular transport	x	
gi|635142324	Ras-related protein Rab-9A	2	0.014	0.4	Intracellular trafficking, secretion, and vesicular transport	x	
gi|635116140	Calumenin	3	0.001	0.5	Metabolism	x	
gi|635014920	nucleobindin-2	2	0.049	0.7	Nuclear protein	x	
gi|635016929	phosphatidylinositol-binding clathrin assembly protein	3	0.013	0.7	Intracellular trafficking, secretion, and vesicular transport	x	
gi|635090518	integrin beta-4	2	0.030	0.8	Signal transduction mechanisms; cell adhesion	x	
gi|635089234	AP-2 complex subunit beta	9	0.031	0.8	Intracellular trafficking, secretion, and vesicular transport	x	
gi|635102142	galectin-1	7	0.047	0.9	Cell cycle control, cell division, chromosome partitioning	x	
gi|635090268	septin-9	4	0.039	0.9	Cell cycle control, cell division, chromosome partitioning	x	
gi|635080465	exportin-1	11	0.049	0.9	Nuclear protein	x	
gi|635015756	26S proteasome non-ATPase regulatory subunit 13	10	0.007	1.1	Postranslational modification; protein turnover		x
gi|635016430	40S ribosomal protein S3	14	0.043	1.1	Cell cycle control, cell division and chromosome partitioning; Translation, ribosomal and biogenesis		x
gi|635023078	T-complex protein 1 subunit theta	23	0.003	1.1	Postranslational modification; protein turnover; chaperones		x
gi|635067296	ubiquitin-conjugating enzyme E2 N/N-like	7	0.011	1.1	Postranslational modification; protein turnover; chaperones;		x
gi|635147631	ADP/ATP translocase 2	7	0.012	1.1	Cell cycle control, cell division, chromosome partitioning		x
gi|635116374	aldose reductase	14	0.047	1.2	Metabolism		x
gi|635068162	sarcoplasmic/endoplasmic reticulum calcium ATPase 2	9	0.002	1.2	Inorganic ion transpot and metabolism		x
gi|635080331	actin-related protein 2	3	0.030	1.2	Cytoskeleton		x
gi|635018272	coatomer subunit delta	7	0.014	1.3	Intracellular trafficking, secretion, and vesicular transport		x
gi|635104202	T-complex protein 1 subunit gamma	20	0.018	1.3	Postranslational modification; protein turnover; chaperones		x
gi|635098341	protein ERGIC-53	5	0.049	1.3	Intracellular trafficking, secretion, and vesicular transport		x
gi|635074604	serine palmitoyltransferase 1	5	0.028	1.4	Metabolism		x

## Data Availability

The data presented in this study are available on request from the corresponding author.

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
