# Peer review of "Tizoxanide Antiviral Activity on Dengue Virus Replication"

_viruses, 2023, doi:10.3390/v15030696_

Round 1
Reviewer 1 Report (Previous Reviewer 3)
No comments
Reviewer 2 Report (Previous Reviewer 1)
After revision, the quality of this paper was improved significantly and can reach the quality standard of this journal. I suggest accepting it without further revision.
This manuscript is a resubmission of an earlier submission. The following is a list of the peer review reports and author responses from that submission.
Round 1
Reviewer 1 Report
The quality of this manuscript was improved significantly after the serious revision by the authors. However, I still think some necessary experiments (also in my previous comments) have to be done before claiming tizoxanide (TIZ) as a promising therapeutic molecule for the treatment of DENV-2 infections. The requested experiments and the reasons are as follows:
1. The antiviral activity of TIZ is required to be tested in another different cells (such as U87 or mosquito cells C6/36). Yes, Vero cells is common for the antiviral research as replied by the authors, however, my concern is TIZ may also target host protein(s) seriously, and it already caused significant toxicity in the current experiment. Testing in another different cell line may rule out the possibility of host cell targeting effects as the different cells may contain different contents or same contents at different levels.
2. The virus induced cytopathic effects (CPE) is required to be evaluated, to validate the therapeutic potential (antiviral activity as well as cellular toxicity). As replied by the authors that CPE usually is evident later in infection, yes, that is also what we want to see as the final therapeutic effects (antiviral effects vs cytotoxicity) of TIZ in dengue virus infection.
3. In Table 2, the authors newly added the SD values. Please check and explain why SD values (3rd column) is bigger than the presenting values (2nd column), which is not acceptable as the too big variation.
Reviewer 2 Report
Thanks for the authors' effors. The revised manuscript looks better but the reviewer cannot recommend this manuscript for publication in VIRUSES due to the following points.
1. Most of DENV-2 infections were performed with the MOI of several hundreds. This is less suitable for natural infections. Does Tizoxanide still work during DENV-2 infection with low MOI? Also, the cell lines derived from human such as Huh7 are recommended to use.
2. Two groups (treated with TIZ or mock in uninfected cells) were missed in Table 3. Although the authors provide several support references, it still remains unclear in terms of the mode of action of Tizoxanide as DENV-2 inhibitor. The manuscript showed that twelve of them with higher abundance in DENV-2 infected Vero cells and 10 proteins increased expression with TIZ treatment after viral infection. What is the conclusion? The functional validation of potential proteins is highly recommended.
Reviewer 3 Report
no comments